# Suitable Patient Selection and Optimal Timing of Treatment for Persistent Air Leak after Lung Resection

**DOI:** 10.3390/jcm13041166

**Published:** 2024-02-19

**Authors:** Yoshikane Yamauchi, Hiroyuki Adachi, Nobumasa Takahashi, Takao Morohoshi, Taketsugu Yamamoto, Makoto Endo, Tsuyoshi Ueno, Tekkan Woo, Yuichi Saito, Noriyoshi Sawabata

**Affiliations:** 1Department of Surgery, Teikyo University School of Medicine, Tokyo 173-8502, Japan; yuichi.saito@med.teikyo-u.ac.jp; 2Department of Thoracic Surgery, Kanagawa Cardiovascular and Respiratory Center, Yokohama 236-0051, Japan; h_adachi_no2@yahoo.co.jp; 3Department of Thoracic Surgery, Saitama Cardiovascular and Respiratory Center, Kumagaya 360-0197, Japan; 4Department of General Thoracic Surgery, Yokosuka Kyosai Hospital, Yokosuka 238-8558, Japan; 5Department of Thoracic Surgery, Yokohama Rosai Hospital, Yokohama 222-0036, Japan; taketsugu61@gmail.com; 6Department of Thoracic Surgery, Yamagata Prefectural Central Hospital, Yamagata 990-2292, Japan; m-endoh@ypch.gr.jp; 7Department of Thoracic Surgery, National Hospital Organization Shikoku Cancer Center, Matsuyama 791-0245, Japan; ueno.tsuyoshi.qz@mail.hosp.go.jp; 8Department of Surgery, Yokohama City University, Yokohama 236-0004, Japan; 9Department of Thoracic and Cardiovascular Surgery, Nara Medical University, Kashihara 634-8521, Japan

**Keywords:** postoperative air leak, risk factor, cumulative distribution, hazard function

## Abstract

Objectives: The choice of therapeutic intervention for postoperative air leak varies between institutions. We aimed to identify the optimal timing and patient criteria for therapeutic intervention in cases of postoperative air leaks after lung resection. Methods: This study utilized data from a prospective multicenter observational study conducted in 2019. Among the 2187 cases in the database, 420 cases with air leaks on postoperative day 1 were identified. The intervention group underwent therapeutic interventions, such as pleurodesis or surgery, while the observation group was monitored without intervention. A comparison between the intervention group and the observation group were analyzed using the cumulative distribution and hazard functions. Results: Forty-six patients (11.0%) were included in the intervention group. The multivariate analysis revealed that low body mass index (*p* = 0.019), partial resection (*p* = 0.010), intraoperative use of fibrin glue (*p* = 0.008), severe air leak on postoperative day 1 (*p* < 0.001), and high forced expiratory volume in 1 s (*p* = 0.021) were significant predictors of the requirement for intervention. The proportion of patients with persistent air leak in the observation group was 20% on postoperative day 5 and 94% on postoperative day 7. The hazard of air leak cessation peaked from postoperative day 3 to postoperative day 7. Conclusions: This research contributes valuable insights into predicting therapeutic interventions for postoperative air leaks and identifies scenarios where spontaneous cessation is probable. A validation through prospective studies is warranted to affirm these findings.

## 1. Introduction

Postoperative air leak is one of the most common complications following lung surgery [1]. Some patients undergoing lung resection may leave the operating room with residual air leak, which usually disappears within the first 24 h. However, 10–20% of patients demonstrate persistent air leak [2]. Empirically, most cases of air leak resolve within the first 5 days postoperatively with conservative chest tube management. In contrast, prolonged air leak (PAL) can lead to a longer duration of hospitalization, delayed physiotherapy and rehabilitation, increased morbidity, and higher healthcare costs, and PAL can also contribute to patient mortality [3,4,5,6]. For these reasons, several preventative strategies for PAL, including surgical techniques, sealants, or buttressing materials, have been tested in previous clinical investigations.

These previous studies can be categorized into two main groups. The first category encompasses studies that have identified risk factors for the development of PAL, and interventional studies that have been conducted in patients with high risk factors. These studies have focused on comparing patients with and without PAL [7,8,9,10,11,12,13]. The second category includes studies that examine methods to stop air leak in cases of postoperative PAL [14,15,16,17,18]. These studies aim to assess the effectiveness of specific approaches to prevent air leak.

Based on these reports, in clinical practice, thorough preoperative and intraoperative assessments are conducted, and various intraoperative strategies are implemented for cases deemed to be at high risk. However, despite numerous studies, several unresolved problems still remain. These unsolved problems are as follows: what characteristics of patients with residual air leaks on postoperative day (POD) 1 are most likely to cease spontaneously, and when they are most likely to cease spontaneously. Essentially, these issues pertain to determining when and under what circumstances clinicians should decide to administer additional therapeutic intervention during the postoperative course. To tackle these questions, we conducted the present study using prospectively collected data from a prior observational study. The study focuses on patients with air leaks on POD 1, aiming to shed light on when and in which cases clinicians should consider additional therapeutic intervention.

## 2. Materials and Methods

### 2.1. Ethical Statement

The institutional review board of our institution approved this study (approval number: 22-038, approval date: 13 July 2022), and the requirement for informed consent was waived due to the retrospective nature of the study. This study was performed in accordance with the principles of the Declaration of Helsinki.

### 2.2. Evaluation Items

-Primary endpoint: Identification of risk factors for PAL development.-Secondary endpoint: Identification of when postoperative air leaks are likely to stop.

### 2.3. Patients

The data analyzed in this study were obtained from a database of prospectively collected data from a previous multicenter observational study, the ILO1805 trial [19]. The ILO1805 trial was conducted as a multi-institutional prospective observational study among 21 Japanese participating institutes for the management of air leak and chest drainage after a lung resection. In the trial, all patients who underwent lung resection for lung tumors in a participating institute were enrolled, except for those who met the exclusion criteria. The exclusion criteria were pneumonectomy, chest wall resection, bronchoplasty, resection for cystic lung disease, resection for infectious lung disease, resection for the purpose of biopsy, and patient refusal to participate.

In the trial, a total of 2200 cases were initially registered; however, 13 cases were excluded from the database due to inadequacy. As a result, this study was conducted on a total of 2187 cases. Indication criteria were patients with confirmed air leak on POD 1. No exclusion criteria were set because the database was sufficiently sophisticated. Comparisons were made between two groups, one in which no additional therapeutic intervention was required for air leak (only continued chest drainage was used; observation [OBS] group), and one in which additional therapeutic intervention was provided for air leak (intervention [INT] group). Surgery and/or pleurodesis were used as the specific interventions in the I group.

To rigorously evaluate the relationship between postoperative air leak and the drainage system in the ILO1805 trial [19], the time taken for air leak cessation and drain removal in increments of 0.5 days were recorded in the database. Therefore, the analyses were also performed at intervals of 0.5 days in this study.

### 2.4. Postoperative Management for Air Leak

The ILO1805 study was conducted as observational research without a predetermined protocol for perioperative management. During lung surgery, one to two chest tubes were inserted, and drainage management was implemented. There were three methods for drainage management: water seal, administered through a three chamber system without external suction; continuous suction, managed by a three chamber system with external suction connected to the patient unit wall or a portable suction pump; and a chest drainage system with digital real-time monitoring of the air leak with continuous suction (Thopaz^®^; Medela Healthcare, Baar, Switzerland). The choice and conversion of the drainage method were based on the surgeon’s preference.

Air leak assessment occurred twice a day (morning and evening) by at least two staff members, including at least one board-certified thoracic surgeon. The cessation of air leak was defined as the absence of bubbles in the water seal chamber even during strong coughing or deep breathing (using conventional methods) or as a mean air leak flow of less than 20 mL/min for at least 4 to 12 h (with digital drainage). The chest tube was removed when the air leak stopped, and the drainage effusion was between 200 and 300 mL per day without indications of blood, chyle, or infection.

### 2.5. Clinicopathological Parameters

Using the information collected in the ILO1805 trial [19], the clinicopathological parameters examined in this study encompassed various factors, including age, sex, preoperative Body Mass Index (BMI), preoperative Forced Expiratory Volume in 1 s (FEV1) percentage, smoking history, presence of interstitial pneumonitis in preoperative imaging assessment, use of corticosteroids, type of lung resection, side of the lung resection, presence of pleural adhesion during lung resection, wound size of lung resection, use of fibrin glue during lung resection, type of interplane cutter used during the lung resection, size of the chest tube used in lung resection, pathological diagnosis of the excised lesion in the lung, and severity of air leak on postoperative day 1 (POD1).

### 2.6. Severity of Air Leak

In line with the definition of the degree of air leak used in the ILO1805 trial [19], air leak severity was classified according to the study by Cerfolio et al. [20]. In patients treated with conventional methods using the three-bottle system, the severity of air leak was categorized as follows: (1) mild (air leak observed only during coughing or forced expiratory phases), (2) moderate (air leak observed in the early expiratory phase but not in the terminal expiratory phase), and (3) severe (air leak observed throughout the expiratory or inspiratory phases, or the inability to achieve complete drainage under continuous suction at a pressure of −10 cmH_2_O). Conversely, in patients treated with digital drainage, the severity of air leak was defined based on previously reported findings [21] as mild (≤100 mL/min), moderate (101–500 mL/min), or severe (≥501 mL/min).

### 2.7. Statistical Analysis

SPSS version 28 (IBM Corporation, Armonk, NY, USA) and GraphPad Prism version 10.0 (GraphPad Prism Software Inc., San Diego, CA, USA) were used for the statistical analyses and to construct the figures. A *p*-value of <0.05 was considered statistically significant. Cumulative distribution function curves were constructed according to the clinical factors, and the curves were compared using the log-rank test. The unpaired *t*-test and the Cox proportional hazards model were used to perform univariate and multivariate analyses to assess the relationships between the clinical factors. In the multivariate analysis, factors were selected according to the results of the univariate analysis.

The hazard function is defined as the instantaneous risk of the event of interest occurring within a fairly narrow timeframe [22]. In this study, the goal of the hazard function was to model a participant’s likelihood of air leak cessation as a function of postoperative time. The cessation rate is herein described as the hazard rate. The time scale was discretized in increments of 0.5 days, and all hazard rates were measured as “events/patients at risk per 0.5-day interval”. The hazard function depended on the number of cases of cessation in a given short period of time [∆t] and the number of cases with air leak within this same short period of time. The hazard function was calculated using the following formula:$$h\;(t)=\frac{Number\;of\;cessation\;cases\;in\;a\;half\;day}{Number\;of\;cases\;air\;leak\;just\;before\;the\;time}$$

Using the results obtained with this formula, a scatterplot was constructed, and the smoothing curve was described using the locally weighted scatter plot smoothing method [23]. The 0.5-day instantaneous hazard rates were estimated because the rate estimates were unstable due to random fluctuation, and a smoothed curve was more useful for understanding the hazard rate patterns.

## 3. Results

There were 420 patients extracted from the database, as shown in Figure 1. Among them, 46 patients underwent intervention (intervention group: INT), whereas 374 patients did not undergo any intervention and were classed as the observation group (OBS).

Table 1 shows the characteristics of the patients and the comparison of the two groups in the univariate analysis in terms of background factors. BMI was significantly lower in the INT group than in the OBS group (*p* = 0.006). The FEV1 percentage was better in INT group than in the OBS group (*p* = 0.054). Regarding the type of surgery, wedge resection was significantly more frequently used in the INT group than in the OBS group (*p* = 0.04). Moreover, pleural adhesion was more frequently found intraoperatively in the INT group than in the OBS group (*p* = 0.052). On the other hand, steroid use and the presence of interstitial pneumonia were not significant factors. The intraoperative use of fibrin glue was significantly more common in the INT group than in the OBS group (*p* = 0.004). In addition, air leak on POD 1 was significantly more severe in the INT group than in the OBS group (*p* < 0.001).

A multivariate analysis was performed to identify the important background factors related to the requirement for intervention in cases of postoperative air leak. In multivariate analysis, we conducted the analysis using factors that were identified as significant in the aforementioned univariate analysis. Table 2 shows the results of the logistic regression analysis to identify predictors of the requirement for intervention for postoperative air leak. The results show that the requirement for intervention was strongly associated with lower BMI, a FEV1 percentage of >80%, wedge resection, intraoperative use of fibrin glue, and severe air leak on POD 1.

Within the INT group, the breakdown of treatment for postoperative air leak (PAL) was as follows: 37 patients underwent chemical pleurodesis (pleurodesis group: PLS) and 9 patients underwent surgery (surgery group: SRG). Table 3 shows a comparison of patients’ characteristics between the PLS group and SRG group. There were no significant differences in the background factors between the PLS group and SRG group, although the thoracic drainage tube inserted at the time of the initial surgery was significantly thinner in the SRG group than in the PLS group (*p* = 0.008).

Air leak on POD 1 was significantly more severe in the SRG group than in the PLS group (*p* = 0.001). The duration to intervention tended to be slightly shorter in the PLS group than in the SRG group, although not significantly different (*p* = 0.11). The number of days from intervention to air leak cessation and from intervention to thoracic drainage tube removal was significantly shorter in the SRG group than in the PLS group (*p* < 0.001 and *p* < 0.001, respectively).

Figure 2 depicts the cumulative distribution function of the air leak cessation rate in both the INT group and the OBS group. Due to the absence of a predetermined perioperative management protocol in the ILO1805 trial, management approaches varied among cases. Consequently, the time required to achieve air leak cessation was found to be longer in the INT group compared to the OBS group (*p* = 0.002). Specifically, in the INT group, air leak cessation was reached in 50% of patients by postoperative day (POD) 10 and in 80% of patients by POD 13. In contrast, within the OBS group, air leak cessation rates were 70%, 80%, and 94% by PODs 4, 5, and 7, respectively.

Here, we specifically focused on the OBS group and measured the hazard function over time to analyze the likelihood of air leak cessation. Figure 3 illustrates the evolution of the hazard function across the postoperative period. As indicated by the smoothing curve, the hazard associated with achieving air leak cessation increased until POD 3, reached its peak on POD 3, and maintained a relatively high level until POD 7. Subsequently, the hazard decreased until approximately POD 9, followed by a gradual and sustained decline thereafter.

## 4. Discussion

The present study examined the characteristics of patients with postoperative air leak on POD 1. We identified the following risk factors for PAL development: low BMI, a FEV1 percentage >80%, wedge resection, use of intraoperative fibrin glue, and severe air leak on POD 1. We also found that postoperative air leak cessation was achieved in 80% of patients by POD 5 without any intervention, and that postoperative air leak cessation tended to be achieved from POD 3 to POD 7. We believe that the identification of when postoperative air leak cessation is likely to be achieved would be important for physicians. Furthermore, the risk factors for PAL among patients with residual air leak on POD 1 are unknown, and the results of this study provide important information for real-world clinical practice.

Regarding the risk factor for PAL development, some of the risk factors identified in the multivariate analysis are consistent with previous reports, whereas others differed from previous reports. First, low BMI was identified as a risk factor in previous reports [8,11]. BMI is thought to act as a risk factor in terms of the effects associated with delayed wound healing at the leak point in patients with poor nutrition [9]. The large amount of air leak as a risk factor is also consistent with previous reports, and its influence is considered to be related to the presence of large leakage points that are unlikely to heal spontaneously.

In contrast to the present study, low pulmonary function, including a low FEV1 percentage, was cited as a risk factor for PAL in previous studies [7,8,9,10,11]. Conversely, this study revealed that better lung function (FEV1 percentage >80%) was a risk factor for PAL. This discrepancy may be related to the fact that this study included only patients with residual air leak on POD 1. It is presumed that patients with better lung function are less likely to experience air leak on POD 1. We suggest that if air leak remained on POD 1 despite good lung function, there must have been a large leak point that did not heal well spontaneously. In contrast to the present study, previous studies have suggested that air leak on POD 1 very rarely exists in patients who undergo wedge resection [7,11]. In the present study, wedge resection was one of the risk factors for PAL. Therefore, if air leak remains despite wedge resection, it is again presumed that there is a large leak point that will not easily heal spontaneously. Furthermore, the current study identified the intraoperative use of fibrin glue as a risk factor for PAL. On the other hand, numerous reports have previously highlighted the efficacy of fibrin glue in stopping air leaks [24,25,26]. In light of this context, this factor can be considered in a manner similar to the previously mentioned factors. In other words, it is suggested that, if air leak persists beyond postoperative day (POD) 1 despite the intraoperative use of fibrin glue, there may be a large leakage point that is unlikely to heal spontaneously.

The definition of PAL in previous studies was continuous air leak varying in duration from 4 to 7 days, with 5 days being the most commonly used definition, despite there being a lack of data upon which to base this definition [6,9,11,12,13]. Based on the analysis of the OBS group in the present study, it is expected that air leak will stop by POD 5 in 80% of patients without the need for additional therapeutic intervention (other than continued chest drainage). This means that therapeutic intervention for air leak should be considered if postoperative air leak is confirmed more than 5 days after surgery. Therefore, it seems reasonable to define PAL as 5 days of air leak continuation.

To answer the question of when therapeutic intervention is needed for PAL, the results of the hazard function analysis in the present study may be helpful (Figure 3). The results show that the probability of air leak cessation without therapeutic intervention was high from POD 3 to around POD 7, after which the probability decreased, reaching two thirds of the peak by POD 9, with a subsequent gradual decrease in probability. In other words, it was presumed that air leak cessation was likely to be achieved without any intervention up to POD 7, whereas cessation was less likely after that time. Furthermore, according to the results of the analysis by cumulative distribution function, among the patients who did not require therapeutic intervention, air leak stopped in 80% by POD 5 and in 94% by POD 7. In view of these results, we suggest that interventions to stop air leak should be implemented in patients in whom air leak has not stopped by POD 7, and that preparation for such interventions should not only begin in cases where air leak has not stopped by POD 5, but it should also be implemented at an earlier date in cases with risk factors that are likely to warrant intervention.

A comparison was made between the PLS group and the SRG group within the INT group to assess differences in intervention methods. Since this study was conducted without a specifically defined perioperative management protocol, it serves to indicate trends in perioperative management methods within this cohort. It was observed that, in cases with a higher volume of air leak, surgery was chosen, resulting in a shorter time for air leak cessation and drainage removal compared to pleurodesis in the surgery group. While performing a re-operation demands substantial medical resources and poses a high implementation hurdle, the efficacy of achieving air leak cessation is considerable. Therefore, considering the higher effectiveness in stopping air leaks, opting for surgery in cases with more severe air leaks is considered reasonable, despite the logistical challenges associated with performing a second surgery.

Regarding drainage management methods, the use of a digital drainage system has emerged as an alternative to traditional approaches. Some studies have highlighted its superiority over conventional drainage methods [27,28]. Nonetheless, there are reports with conflicting findings, suggesting that the digital drainage system does not necessarily reduce the duration of chest drainage or hospital stays [29,30]. Furthermore, the ILO1805 study, which contributed data to the research database utilized in this study, concluded that water seal management was superior to the digital drainage system [19]. These discrepancies may stem from a lack of consensus on the optimal utilization of digital drainage systems. Our study findings indicate that postoperative air leaks are more likely to cease between POD 3 and POD 7. It is hypothesized that providing milder suction pressure with the digital drainage system around this timeframe may facilitate the cessation of air leaks more effectively.

This study has several notable limitations. First, this study was retrospectively conducted using prospectively collected data from another research database. Due to the different objectives of the original study, there are limitations in terms of the data that were collected, and it was not possible to cover everything with regard to the items that had been previously identified as risk factors. Additionally, information on potentially relevant factors, such as a visual evaluation method for radiologic emphysema and the duration of hospital stay, has not been collected. Second, the original multicenter prospective observational study did not have a uniform policy on drain management strategies, leaving it up to the individual facilities to decide the strategy for drain management. Therefore, the results obtained may have been influenced by the policy of the institutions that recruited the largest patient numbers. However, the results of the questionnaire survey of the individual facilities indicated that a relatively equal number of facilities were proactive and reactive toward PAL, suggesting that the results may have bias. For similar reasons, it is difficult to solve the question of how to choose between pleurodesis and surgery as a treatment for postoperative PAL. However, the use of original data from a multicenter prospective setting is a strength of this research.

## 5. Conclusions

This study identified predictors of the requirement for therapeutic intervention for postoperative PAL in patients with air leaks on POD 1. In addition, the time taken to achieve air leak cessation without additional intervention was identified. Moreover, it seems reasonable to define PAL as air leak continuing for more than 5 days after surgery. Prospective studies are needed to validate the results.

## Figures and Tables

**Figure 1 jcm-13-01166-f001:**
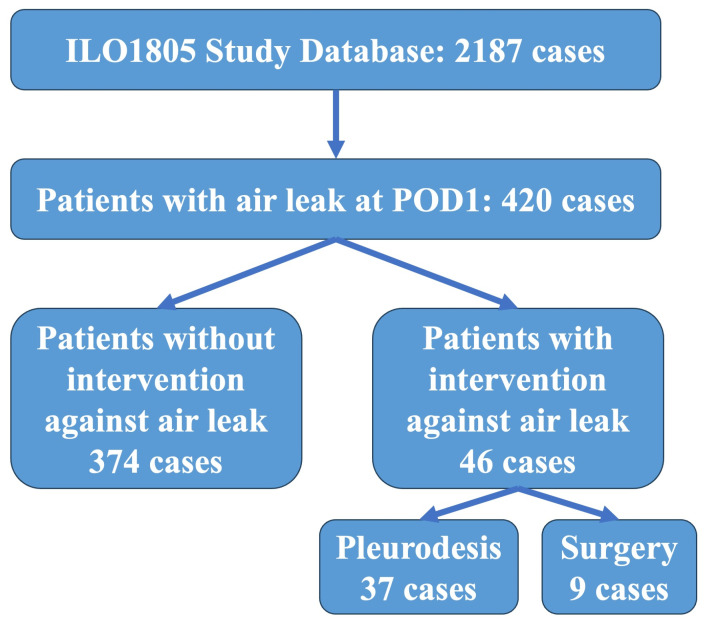
Flowchart of patient selection and classification.

**Figure 2 jcm-13-01166-f002:**
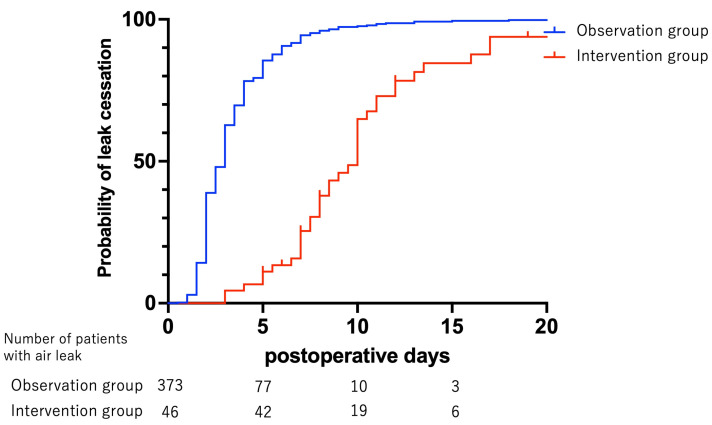
The cumulative distribution function of the air leak cessation rate in the intervention and observation groups. The red line and blue line indicate patients in the intervention and observation groups, respectively. The intervention group had a longer overall time to air leak cessation than the observation group (*p* = 0.002).

**Figure 3 jcm-13-01166-f003:**
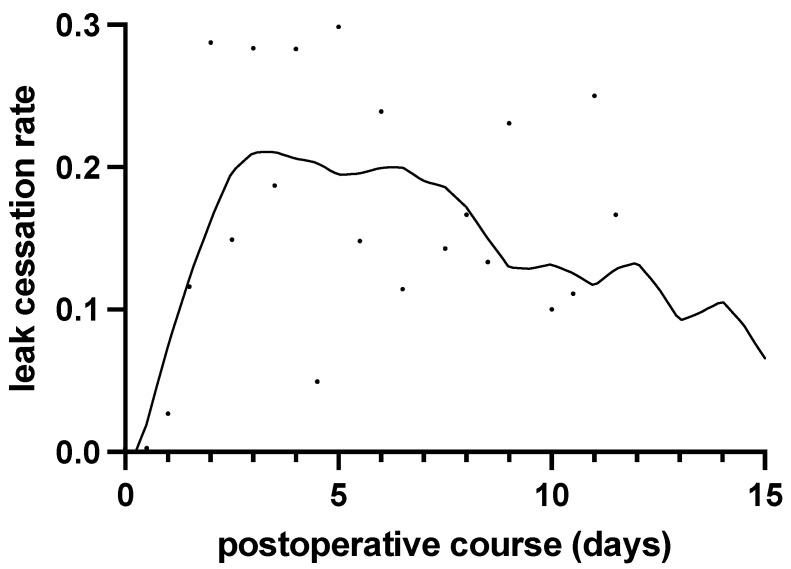
The transition of the hazard function of air leak cessation with postoperative period. The solid line shows the smoothing curve described with the locally weighted scatter plot smoothing method. The hazard at which air leak stopped increased until postoperative day 3, peaked on postoperative day 3, and remained high until postoperative day 7. After that, the hazard decreased until around postoperative day 9 and then slowly decreased thereafter.

**Table 1 jcm-13-01166-t001:** Characteristics of the patients in the intervention and observation groups and the univariate analysis of the requirement for intervention in patients with postoperative air leak.

Factors		Intervention Group (INT, *n* = 46)	Observation Group (OBS, *n* = 374)	*p*-Value
Age		72 ± 9 (39–88)	71 ± 9 (19–89)	0.550
Sex	Male	37 (80%)	262 (70%)	0.142
	Female	9 (20%)	112 (30%)	
Body mass index (kg/m^2^)		21.2 ± 3.4 (12.3–27.8)	22.7 ± 3.5 (14.8–35.3)	0.006
Smoking (pack × year)		39 ± 31 (0–108)	33 ± 32 (0–190)	0.216
FEV1.0%	<70%	12 (26%)	127 (34%)	0.054
	70–80%	14 (30%)	148 (40%)	
	>80%	20 (43%)	99 (26%)	
Interstitial pneumonitis (case)		1 (2%)	45 (12%)	0.297
Corticosteroids (case)		1 (2%)	9 (2%)	0.922
Surgery type	Lobectomy/Segmentectomy	33 (72%)	314 (84%)	0.039
	Wedge resection	13 (28%)	60 (16%)	
Surgery side	Left	20 (43%)	135 (36%)	0.328
	Right	26 (57%)	239 (64%)	
Pleural adhesion		19 (41%)	103 (28%)	0.052
Wound size	Small	33 (72%)	244 (65%)	0.631
	Medium	4 (9%)	33 (9%)	
	Large	9 (20%)	97 (26%)	
Use of Fibrin glue		39 (85%)	238 (64%)	0.004
Inter-plane cutter	Energy device	1 (2%)	21 (6%)	0.323
	Stapler	45 (98%)	353 (94%)	
Drain size (Fr)		22 ± 2 (18–24)	21 ± 2 (16–28)	0.104
Pathological Diagnosis	Benign tumor	0 (0%)	1 (0.2%)	0.717
	Inflammation	0 (0%)	11 (3%)	
	Lung cancer	42 (91%)	323 (86%)	
	Metastasis	4 (9%)	35 (9%)	
	Others	0 (0%)	4 (1%)	
Air leak on POD 1	Mild	20 (43%)	311 (83%)	<0.001
	Moderate	21 (46%)	60 (16%)	
	Severe	5 (11%)	3 (1%)	

FEV1%: percent predicted forced expiratory volume in 1 s, POD 1: postoperative day 1.

**Table 2 jcm-13-01166-t002:** Results of the multivariate logistic regression analysis of the requirement for additional intervention in patients with postoperative air leak.

Factors	*p*-Value	Odds Ratio	95% Confidence Interval
Body mass index (kg/m^2^)	0.019	0.879	0.79–0.979
FEV1.0% (>80%)	0.021	1.69	1.084–2.634
Surgery type (Lobectomy/Segmentectomy)	0.01	0.329	0.141–0.77
Pleural adhesion	0.218	1.573	0.765–3.234
Use of fibrin glue	0.008	3.456	1.384–8.629
Air leak on POD 1 (severe)	<0.001	5.374	2.981–9.689

FEV1 percentage: percent predicted forced expiratory volume in 1 s, POD 1: postoperative day 1.

**Table 3 jcm-13-01166-t003:** Characteristics of the patients who underwent pleurodesis or surgery as the first intervention for postoperative air leak.

Factors		Pleurodesis Group (PLS, *n* = 37)	Surgery Group (SRG, *n* = 9)	*p* Value
Age		73 ± 8 (49–88)	68 ± 13 (39–77)	0.39
Sex	Male	30 (81%)	7 (78%)	0.82
	Female	7 (19%)	2 (22%)	
Body mass index (kg/m^2^)		21.4 ± 3.5 (12.3–27.8)	20.2 ± 3.8 (16.3–26.7)	0.14
Smoking (pack × year)		41 ± 32 (0–108)	31 ± 23 (0–64.5)	0.62
FEV1.0%	<70%	8 (22%)	4 (44%)	0.27
	70–80%	11 (30%)	3 (33%)	
	>80%	18 (49%)	2 (22%)	
Surgery type	Lobectomy/Segmentectomy	28 (76%)	5 (56%)	0.23
	Wedge resection	9 (24%)	4 (44%)	
Wound size	Small	27 (73%)	6 (67%)	0.93
	Medium	3 (8%)	1 (11%)	
	Large	7 (19%)	2 (22%)	
Drain size (Fr)		22 ± 2 (18–24)	20 ± 2 (18–24)	0.008
Air leak on POD 1	Mild	18 (49%)	2 (22%)	0.001
	Moderate	18 (49%)	3 (33%)	
	Severe	1 (2%)	4 (44%)	
Duration from initial surgery to intervention (days)		5.3 ± 2.2 (2–11)	7.4 ± 4.1 (1–15)	0.11
Duration from intervention to air leak cessation (days)		3.8 ± 2.5 (0–9.5)	0.3 ± 0.7 (0–2)	<0.001
Duration from intervention to chest drain removal (days)		5.5 ± 3.1 (1–14.5)	0.6 ± 1.3 (0–3.5)	<0.001

BMI: body mass index, FEV1. percentage: percent predicted forced expiratory volume in 1 s, POD 1: postoperative day 1.

## Data Availability

The data underlying this article will be shared on reasonable request to the corresponding author.

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
