# Peer review of "Suitable Patient Selection and Optimal Timing of Treatment for Persistent Air Leak after Lung Resection"

_jcm, 2024, doi:10.3390/jcm13041166_

Round 1

Reviewer 1 Report

Comments and Suggestions for Authors

I would like to congratulate you on your study. You have used the available data to develop and answer a new question. 

Overall, you have a very low air leakage (20%) at POD1. 

However, I did not find any data on the type of surgical approach (open/minimally invasive). Only the size of the wound. What does this refer to? 

Only a fraction of these patients were treated (10%), and for this purpose only 2% were treated surgically. This is a good result. But, I have not seen the impact on the length of stay. The majority of surgeons would be proactive to shortening the hospital stay.

The conclusion - regarding the 5 days limit to call a PAL is derived from your data. Why do you still mention days 3 and 7?

Best regards, and all the best with your Manuscript.

Author Response

Here are the point-by-point reply for reviewer's comments.

Q1) Overall, you have a very low air leakage (20%) at POD1. 
However, I did not find any data on the type of surgical approach (open/minimally invasive). Only the size of the wound. What does this refer to? 

A1) I am grateful for Reviewer 1's insightful comments. Our database did not include information on the performance of thoracoscopic surgery. The rationale behind this omission stems from the context of this study's execution across numerous Japanese facilities, where thoracoscopic surgery is the prevalent method for most cases. Consequently, we posited that wound size serves as the sole viable metric to evaluate surgical invasiveness. As such, cases identified with a small wound size were categorized under the umbrella of "minimally invasive surgery." 

Q2) Only a fraction of these patients were treated (10%), and for this purpose only 2% were treated surgically. This is a good result. But, I have not seen the impact on the length of stay. The majority of surgeons would be proactive to shortening the hospital stay.

A2) Our database did not collect information on the duration of hospital stays. The background to this is that the length of hospital stays can be extended by social factors unrelated to air leak prolongation, making evaluation difficult. This point has been added to the limitations.

Q3) The conclusion - regarding the 5 days limit to call a PAL is derived from your data. Why do you still mention days 3 and 7?

A3) The information that postoperative air leaks are more likely to cease between day 3 and day 7 is considered valuable for physicians managing perioperative care. This point has been added to the discussion in the manuscript.

Reviewer 2 Report

Comments and Suggestions for Authors

I congratulate the Authors for this paper on "...optimal timing of treatment for persistent air leak after lung resection". The study aims to identify risk factors associated with prolonged air leaks after lung resection and to analyze the timing and characteristics of treatments. The number of patients considered in the paper is adequate to come to final consideration. Limitation are already discussed by the Authors and following their considerations, I have a couple of comments about the paper:

1) Regarding the "Clinicopathological Parameters" of patients, did the Authors consider a visual evaluation method of radiologic emphysema using computed tomography (i.e. the Goddard score)? I understand that this is a multicenter retrospective study and there are differences in the data collection.

2) I agree that one of the notable limitations is the nonuniform policy on drain management strategies. Differences in air leaks using digital drainage have been already reported in the literature and, the Authors should mention that in the Discussion section.

Author Response

Here are the point-by-point reply for reviewer's comments.

Q1) Regarding the "Clinicopathological Parameters" of patients, did the Authors consider a visual evaluation method of radiologic emphysema using computed tomography (i.e. the Goddard score)? I understand that this is a multicenter retrospective study and there are differences in the data collection.

A1) I am grateful for Reviewer 2's insightful comments. Our database does not assess a visual evaluation method of radiologic emphysema. The absence of this assessment has been acknowledged as a limitation in our study.

Q2) I agree that one of the notable limitations is the nonuniform policy on drain management strategies. Differences in air leaks using digital drainage have been already reported in the literature and, the Authors should mention that in the Discussion section.

A2) The use of a digital drainage system for air leak management was included in the discussion.